# Proximity Effect of Optically Active h-BCN Nanoflakes Deposited on Different Substrates to Tailor Electronic, Spintronic, and Optoelectronic Properties

**DOI:** 10.3390/ijms26052096

**Published:** 2025-02-27

**Authors:** Ahmad Alsaad, Jaeil Bai, Wai-Ning Mei, Joel Turallo, Carolina Ilie, Renat Sabirianov

**Affiliations:** 1Department of Physics, University of Nebraska at Omaha, Omaha, NE 68182, USA; jibai777@gmail.com (J.B.); physmei@unomaha.edu (W.-N.M.); rsabirianov@unomaha.edu (R.S.); 2Department of Physics, Jordan University of Science and Technology, P.O. Box 3030, Irbid 22110, Jordan; 3Department of Physics and Astronomy, State University of New York, Oswego, NY 13126, USAcarolina.ilie@oswego.edu (C.I.)

**Keywords:** proximity effect, optically active, h-BCN nanoflakes, chiral-induced spin selectivity, chirality, electronic, spintronic, optoelectronic, bandstructure, time-reversal invariance

## Abstract

Hexagonal BCN (h-BCN), an isoelectronic counterpart to graphene, exhibits chirality and offers the distinct advantage of optical activity in the vacuum ultraviolet (VUV) region, characterized by significantly higher wavelengths compared to graphene nanoflakes. h-BCN possesses a wide bandgap and demonstrates desirable semiconducting properties. In this study, we employ Density Functional Theory (DFT) calculations to investigate the proximity effects of adsorbed h-BCN flakes on two-dimensional (2D) substrates. The chosen substrates encompass monolayers of 3D transition metals and WSe_2_, as well as a bilayer consisting of WSe_2_/Ni. Notably, the hydrogen-terminated h-BCN nanoflakes retain their planar configuration following adsorption. We observe a strong interaction between h-BCN and fcc-based monolayers such as Ni(111), resulting in the closure of the optical bandgap, while the adsorption energy on WSe_2_ is significantly weaker, preserving an approximate 1.1 eV bandgap. Furthermore, we demonstrate the magnetism induced by the proximity of adsorbed chiral h-BCN molecules, and the chiral-induced spin selectivity within the proposed systems.

## 1. Introduction

The effects of symmetry at the interface of two materials represents a potent approach in the search for new phenomena in physics [1,2,3]. This is particularly true in spintronic applications where the spin state of either constituent can strongly affect the properties of the layered structures [4,5,6,7]. Recently, the effect of molecular symmetry has revealed several interesting effects. Molecular spintronics have demonstrated that *chiral* molecules act as spin filters; this effect has been coined “chiral-induced spin selectivity” (CISS). [7,8,9,10,11]. This effect violates the Onsager relation protected by time-reversal invariance that implies that *G*(**M**) = *G*(-**M**) [12,13]. The reciprocity theorem does not apply if time-reversal invariance is broken. Hence, one might expect that aligning two helical molecules in series with a small resistor in between will yield different results, depending on whether the molecules have the same or opposite helicities [14]. The chirality of the molecules, defined by their inability to be superimposed onto their mirror images, gives rise to their unique optical properties. Chirality can influence the transmission of unbound photoexcited electrons, as well as the transport of bound electrons, between leads, through a chiral medium [15]. There are even examples of magnetization switching due to adsorbed chiral molecules [16]. In addition to spin-related phenomena, chirality also plays a significant role in chemical reactions. However, despite growing interest in molecular chirality effects at the interfaces, the understanding of the interfacial properties is still lacking. Exploring these interfacial effects is relevant not only to fundamental research but also to technological innovations. For example, tailoring the chirality of quantum dots could lead to the design of stable and size- and shape-controllable *optically active* systems. Such advances are important for the development of next-generation electronic devices and could provide a pathway to qubit design for quantum computing [17,18,19].

The h-BCN with even contents of boron/carbon/nitrogen is “isolectronic” to graphene, meaning that it may exhibit many of the favorable phenomena that have been discovered in graphene [20,21,22]. However, there are certain differences from graphene that make h-BCN promising for various applications. The h-BCN monolayer typically maintains inversion symmetry. However, a chiral domain can be created by structural modifications [23]. As a result, h-BCN possesses the ability to produce circularly polarized light, which has its own unique applications [24,25]. It has recently been reported that h-BCN is optically active near VUV at higher wavelengths than graphene, has a bandgap that typically lies between 1 and 3.4 eV, and provides a good middle ground between graphene and h-BN [26,27,28].

The small chiral h-BCN molecule can be synthesized and can serve as an ideal system for investigating symmetry-related effects both for the free-standing molecule and for a molecule deposited on a substrate [29,30]. In this study, we performed a DFT-based study on the chiral h-BCN nanoflake, focusing on its electronic and optical properties, with particular emphasis on the interaction with the substrate. Two-dimensional (2D) nanomaterials, particularly anisotropic mono-elemental semiconductors, have attracted considerable attention for optoelectronic and electronic applications [31,32]. We explored various substrate types, including metallic and semiconducting 2D transition metal dichalcogenide (TMD) materials, and investigated the possibility of decorating h-BCN using magnetic 3D atoms. Specifically, we examined the role of chirality in influencing magnetic interactions and magnetic structure within this simple yet promising structure.

## 2. Methodology

The Vienna Ab initio Simulation Package (VASP6.4) was utilized with the Perdew–Burke–Ernzerhof (PBE) generalized gradient approximation (GGA) functional, augmented by the Grimme3 set of empirical dispersion corrections (PBE + D3). The energy cutoff was set at either 500 eV, for the h-BCN/Ir(111) and h-BCN/WSe_2_ systems, or 600 eV, for the isolated h-CBN, h-BN, and graphite structures [33,34,35]. A dipole correction was applied perpendicular to the xy-plane in the slab models.

For Heisenberg exchange and Dzyaloshinskii–Moriya interaction (DMI) calculations for the BCN flake with doping magnetic ions, we used OpenMX code [36] and the Tight Binding to Jastrow (TB2J) method. The OpenMX generated the Hamiltonian and the overlap matrices of the system, which were subsequently used by the TB2J code to calculate the Heisenberg exchange interaction and DMI parameters between localized magnetic moments (spins) in the system [37]. For OpenMX3.9 DFT calculations, the following computational parameters were selected. The GGA-PBE functional [38] and the Kohn–Sham equations were solved using pseudoatomic orbitals (PAOs) as the basis functions [11]. PAOs for the atoms involved were taken as follows: B^7^.^0-^s^2^p^2^d^1^, C^6^.^0-^s^2^p^2^d^2^, N^6^.^0-^s^2^p^2^d^1^, H^6^.^0-^s^2^p^1^, Co^6^.^0^H^-^s^3^p^2^d^1^, and Ni^6^.^0^H^-^s^3^p^2^d^1^. The cutoff energy was 150 Rydberg. The SCF convergence criterion for TB2J input was 1.0 × 10^−8^. The convergence criterion for geometry optimizations was established at 0.001 eV/Å for force calculations. We performed the full geometry of the system under consideration with forces 10^−2^ eV/Å. The adsorption energy was calculated as the difference between the total energy of the combined system (h-BCN flake + substrate) and the sum of the individual energies of the isolated h-BCN flake and the substrate. Mathematically, it can be expressed as follows:$$E_{ads}=E_{h-BCN/substrate}-(E_{h-BCN}+E_{substrate})$$

A negative value of $E_{ads}$ indicates a stable adsorption, with a stronger interaction corresponding to more negative values. We performed DFT with spin–orbit coupling and Hubbard U corrections for TM d-states. Hubbard U was chosen as U(Mn) = 3 eV, U(Co) = 3 eV, and U(Ni) = 3 eV.

DFT calculations of the h-BCN molecule decorated by magnetic atoms were performed by OpenMX [39] code, followed by TB2J calculations of magnetic exchange interaction coupling parameters between localized magnetic moments (spins) in the system. We used wannier90 [40] and TB2J [37] for calculations of exchange interactions. The magnetocrystalline anisotropy energy was calculated by the difference in energies of in-plane and out-of-plane spin arrangements using the following force theorem:$$E_{MAE}=E_{↑}-E_{\to }$$

## 3. Results and Discussion

### 3.1. Properties of h-BCN Molecule on Metallic Substrate

Figure 1 shows the atomic structure of the h-BCN decorated by hydrogen atoms. This flake is clearly chiral, about 1.2 nm in diameter, and expected to be stable. DFT calculations performed to analyze its electronic structure show that the flake has a HOMO-LUMO gap of about 1.6 eV (Figure 1c). PBE approximation used to treat the exchange-correlation potential (the variety of generalized gradient approximation) underestimates the bandgap by 30–40%. However, it is still believed to be a promising electronic material with optical activity.

For practical device applications, the h-BCN flake needs to be deposited on the substrate. We investigated the interaction of the flake with the metallic substrate, specifically Cu, Ir, and Ni. Copper and iridium are common conducting materials, while nickel is not only magnetic but also commonly used in the synthesis of graphene [15]. Figure 1b shows the relaxed structure of the h-BCN flake on Cu. Notably, the flake undergoes significant corrugation, losing its planar structure, which indicates strong bonding to the substrate. This observation is further supported by the analysis of bonding in Table 1, where the adsorption energy is of the order of ~10 eV, indicating that a robust bonding is established with the substrate.

However, we attribute such strong bonding to the strong hybridization of the flake’s electronic states with those of the substrate. This interaction has a profound effect on the electronic states of the system. As shown in the band structure of the supercell containing the slab and the nanoflake (Figure 1d), this strong hybridization effectively closes the optical gap and renders it difficult to apply the material in optoelectronics.

### 3.2. Proximity Effect in h-BCN/WSe_2_ Bi-Layer

To address the drawback of strong hybridization while harnessing the benefits of symmetry breaking, we selected WSe_2_ as the preferred substrate. WSe2 is a monolayer 2D transition metal dichalcogenide with a similar bandgap to h-BCN, known for its proximity effect and distinct splitting of right- and left-polarized light when interacting with magnetic substrates like EuS. The strong spin–orbit coupling in WSe_2_ results in opposite spin splitting at positive and negative momentum k values, which can be advantageous for spintronic/valleytronic applications.

The DFT analysis of the h-BCN on WSe_2_, shown in Figure 2a, indicates that the interaction between h-BCN and WSe2 in the bilayer configuration is weak, with a low adsorption energy. This results in a type I band alignment, where both conduction and valence band edges align within the same material, reducing the combined h-BCN/WSe_2_ bandgap to 1.1 eV (from intrinsic bandgaps of 1.6 eV for h-BCN and 1.23 eV for WSe_2_). Despite the weak interaction, a proximity effect is clearly observed in the bilayer system shown in Figure 2.

Although h-BCN itself has a zero net spin moment, spin character is induced in the states with opposite k-directions. This effect is obviously connected with the spin–orbit coupling in WSe_2_ and the proximity effect. Additionally, there is a noticeable change in the degree of spin polarization in WSe_2_, as indicated by the color intensity variations in Figure 2b,c. The red arrows at the K’ points show that the valence band edge state splits in the presence of h-BCN and its spin polarization $\left(N^{↑}-N^{↓})/(N^{↑}+N^{↓}\right)$ also decreases from ~60% to 44%. These changes can affect the spin-selective optical excitations of the bilayer.

### 3.3. Interfacial Effect in h-BCN Deposited on Defect-Modified WSe_2_ Monolayer

The introduction of defects in WSe_2_ readily changes the symmetry of the system. When magnetic ions are introduced, such as Mn, either as a substitutional dopant or intercalating site, the time-reversal symmetry breaks. For the geometries shown in Figure 3a,b, we find that the bands associated with WSe_2_ experience a k-dependent shift in band dispersions. As can be seen in Figure 3c, the Mn impurity on the W site dispersions along Γ-K and Γ-K’ are asymmetric. Thus, the PL spectra of such a system will have splitting related to left and right circularly polarized light. At the same time, the Mn-impurity state does not have spin dependence related to opposite k-vector directions due to the time-reversal symmetry breaking. This is also true for h-BCN states, with all five impurity states in the bandgap having some spin character but not showing significant momentum directional asymmetry. A similar observation can be made in the case of the Mn atom being intercalated between h-BCN and WSe_2_, as shown in Figure 3f. The main dispersion asymmetry for WSe_2_ bands is observed in the conduction band.

Modifying the WSe_2_ monolayer through doping or irradiation can significantly influence the interaction strength between the substrate and the h-BCN molecule. For instance, electron irradiation can create vacancies within the WSe_2_ lattice. Given that Se atoms have a lower atomic number than W, they are more susceptible to ejection under electron irradiation, leading to Se vacancies that alter the material’s electronic and chemical properties at the interface [41,42].

We considered an h-BCN molecule deposited on a WSe_2_ substrate with single selenium (Se) ion vacancies, as well as two and four selenium ion vacancies, in the WSe_2_ monolayer. As can be seen in Figure 3d, the two defect states appear in the bandgap when a single Se vacancy is created ~0.3–0.5 eV below the conduction band of WSe_2_. These states have opposite spin characters and reflect time-reversal symmetry in the presence of SO interactions. The n- and p- doping can be achieved using impurities such as As and Br. Figure 3e shows that As-doped WSe_2_ as a substrate develops a hole pocket at K and K’ points of opposite spin character. The band alignment of h-BCN and *W*${Se}_{2}$ is perfect in this case, and we do not observe any impurity states in the bandgap. The bandgap is dominated by WSe_2_ states in this case. The Br doping of WSe2 causes electron doping (contrary to the As case), as can be seen in Figure 3h. The impurity state is hybridizing with the conduction band reflected in its spin character (there is also a small dispersion due to the finite size of the supercell). The bandgap decreases to 1 eV and is dominated by the h-BCN states. Charge transfer to the h-BCN molecule is possible with these substrate modifications.

The Se ion defects in WSe_2_ introduce localized states within the bandgap. These defect states can act as trap states for carriers, affecting the overall conductivity and optical properties of the material. The presence of defects might also lead to localized strain, which can further modify the band structure by altering the electronic states in the vicinity of the defect. Strain induced by placing the h-BCN flake on WSe_2_ can further modify the band structure. Strain can cause shifts in the conduction and valence bands, leading to a change in the bandgap and effective mass of the carriers. This modification is crucial for potential applications in optoelectronics or spintronics.

We next investigated the h-BCN/WSe_2_ after doping extra transition metals, Mn, Ni, and Co on WSe_2_. The transitions metals Mn, Ni, and Co are stabilized on the top of the W ion site of WSe_2_ and near the C-N bond of the BCN flake. The distances between ions in structures containing TM adsorbed on WSe_2_, shown in Table 2, indicate the establishment of relatively strong bonding. Due to interactions with h-BCN, the on-site exchange splitting of Mn ion is different from the case of substitutional Mn impurity, as can be clearly seen through comparison with Figure 3c,f. It also causes an electron population in one of the h-BCN states just above the WSe_2_ valence edge, indicating a charge transfer from Mn to h-BCN. This strong coupling is reminiscent of one observed in the case of the h-BCN flake deposited on metal surfaces. Mn states are spin-polarized, with a spin moment of ~3.12 µB (compared to 1.35 µB in the case of substitution for W), and this polarizes the states of h-BCN. Similar observations can be made in Co ion intercalation case. The magnetic moment of the adsorbed TM ion in between the BCN flake and WSe_2_ is 0.76 µB for Co. We find that intercalated Ni shows non-magnetic behavior. As a result, it does not break the time-reversal symmetry and WSe_2_ bands are symmetric in Γ-K and Γ-K’ directions, as can be seen in Figure 3g. However, it does affect the h-BCN molecular orbitals both in terms of its relative energies and, specifically, the spin character. It becomes more complex, as can be seen from the color coding of spin states in Figure 3. The effect of interactions of magnetic atoms with the flake was examined using the hexamer decoration of h-BCN with the TM atoms. We found that the overall magnetic structure is sensitive to the specific TM, i.e., the population of the d-shell. The ground state of the Co and Ni hexamer is antiferromagnetic. The symmetry of the system decreases. It is reflected particularly strongly in DMI interactions lacking hexagonal symmetry, as shown in Appendix A.

Finally, depending on the TM, the anisotropy of the system governs the magnetic moment of the antisymmetric system. Specifically, we observed in-plane magnetization for Co and out-of-plane magnetization for Ni. The magnetocrystalline anisotropy energy (MAE) is −0.008 and +0.003 eV, respectively.

## 4. Conclusions

In conclusion, we investigated the electronic structure and magnetic interactions of the chiral h-BCN molecule deposited on substrates ranging from insulating (WSe_2_) to metallic transition metals (Ni, Cu, and Ir). Our findings demonstrate that the nature of the substrate significantly influences the properties of the molecule. Metallic substrates form strong bonds with h-BCN, leading to the closure of its optical gap, while weaker bonding on WSe_2_ preserves the optical gap at ~1.1 eV. Most importantly, the proximity effect induces modifications in the spin character of both the h-BCN molecule and WSe_2_ substrate, highlighting the interplay between molecular chirality and substrate interactions.

Furthermore, we observed that defect modifications and doping strategies, such as n- and p-doping with *non-magnetic defects*, can tune the electronic properties of the system, preserving the spin character of h-BCN states induced by the proximity. In contrast, doping with *magnetic 3D transition metal* atoms disrupts time-reversal symmetry, which is reflected in h-BCN electronic states.

Finally, for h-BCN decorated with multiple 3D adatoms that form a hexamer, our analysis reveals that the Dzyaloshinskii–Moriya interactions are highly sensitive to the symmetry at the interface. In the cases of Co and Ni hexamers that we studied, the Heisenberg exchange interactions were strong, and only minimal magnetic non-collinearity due to DMI was observed. We believe that these findings provide valuable insights into the tunability of chiral molecular systems and their potential applications in spintronics and molecular electronics.

## Figures and Tables

**Figure 1 ijms-26-02096-f001:**
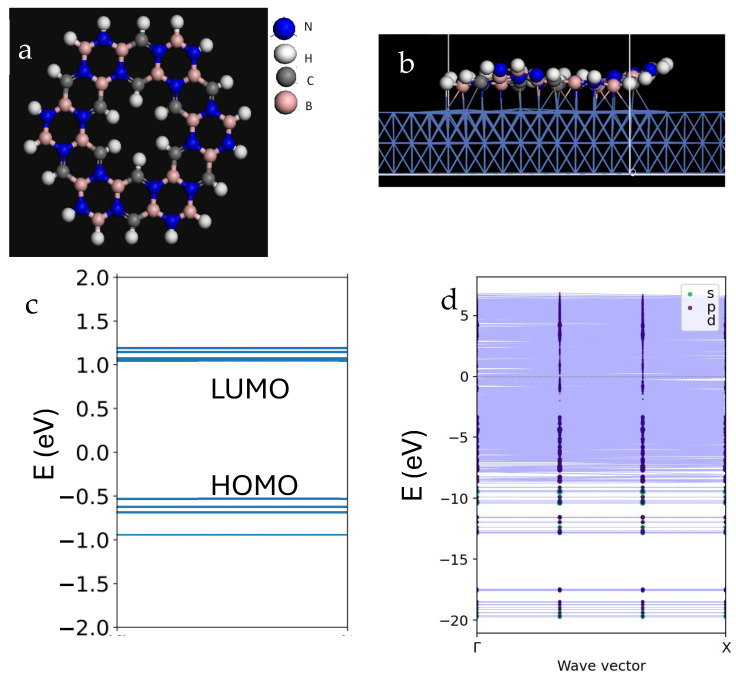
(**a**) Atomistic structure of chiral h-BCN flake. (**b**) Atomistic structure of h-BCN flake on Cu substrate. Two-dimensional unit cell slab with 3 Cu layers is used to represent the Cu (111) surface. (**c**) Molecular orbital diagram showing relative MO energies. (**d**) Band dispersions along G-X direction. The dots show the contribution of states from constituents of the h-BCN flake. Clearly, the states are hybridized with Cu states and show no bandgap, because Cu is a metal.

**Figure 2 ijms-26-02096-f002:**
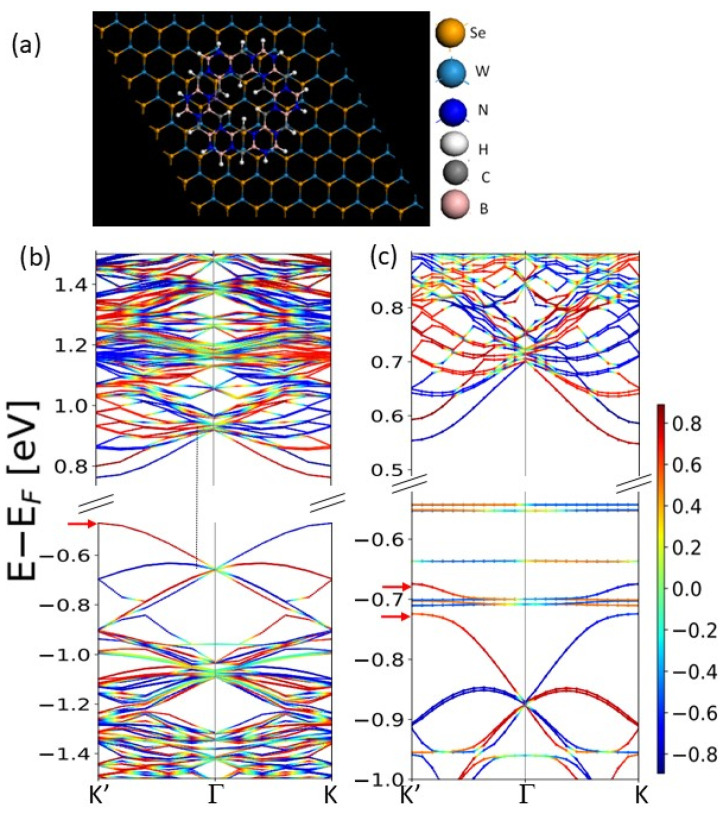
(**a**) Top view (also need side view). The 8 × 8 supercell of WSe_2_ is used to model the substrate. (**b**) Spin-resolved band structure of monolayer WSe_2_ and, (**c**) bilayer h-BCN/WSe_2_. The color bar shows the spin fraction of the state. It is notable that in k and -k, the spin character of the top valence band and lowest conduction band are opposite. Red arrows show the edge of the conductance band.

**Figure 3 ijms-26-02096-f003:**
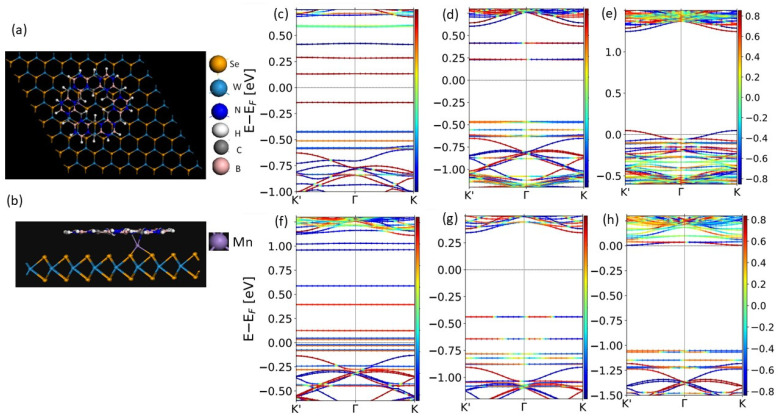
Effect of the symmetry of the bilayer on its band structure. The supercell of the h-BCN flake on the WSe_2_ doped with a single Mn substituting W (**a**) and containing an intercalated Mn atom (**b**). Band structure of the supercell containing (**c**) a single Mn ion substituting W in the WSe_2_ layer, (**d**) a Se vacancy in WSe_2_ layer, (**e**) an As ion substituting W in the WSe_2_ layer (hole doping), (**f**,**g**) Mn and Ni intercalated between the h-BCN and WSe2 layer, respectively, and (**h**) a single W ion replaced with Br (electron doping).

**Table 1 ijms-26-02096-t001:** Adsorption energy of the nanoflake on Cu substrate modeled by two-layer (2L) and three-layer (3L) slabs.

Substrate (10 × 10xnL)	Ir (2 L)	Ir (3 L)	Ni (2 L)	Ni (3 L)	Cu (2 L)	Cu (3 L)
Adsorption energy, eV	−6.58	−9.33	−9.05	−11.36	−0.88	−0.73

**Table 2 ijms-26-02096-t002:** Interatomic distances between sites of h-BCN and WSe_2_ and the intercalated TM ions: Mn, Co, and Ni (in Å).

	Mn	Co	Ni
N-TM	2.03	2.112	2.112
C-TM	2.11	2.125	2.125
Se-TM	2.49, 2.46	2.44, 2.32	2.35

## Data Availability

Data will be made available on request.

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
