# Peer review of "Proximity Effect of Optically Active h-BCN Nanoflakes Deposited on Different Substrates to Tailor Electronic, Spintronic, and Optoelectronic Properties"

_ijms, 2025, doi:10.3390/ijms26052096_

Round 1
Reviewer 1 Report
Comments and Suggestions for Authors
In this manuscript, the authors utilized Density Functional Theory (DFT) calculations to investigate the proximity effects of optically active h-BCN nanoflakes. The theoretical analysis provides a clear understanding of the electronic structure and magnetic interactions in chiral h-BCN molecules deposited on various two-dimensional substrates. And this manuscript is coherent in writing. Therefore, I recommend its publication in International Journal of Molecular Sciences after revision.
1. The authors should provide a clear explanation for selecting monolayers of 3d transition metals, WSe2, and the WSe2/Ni bilayer as substrates for the DFT calculations.
2. The layout of the figure should be adjusted to enhance clarity and comprehension. What’s worse, the content of some figure has been cropped.
3. This manuscript contains a few typos. Please proofread carefully. Examples, the subscripts of WSe2 should be modified.
4. Some of the following references would enrich the manuscript and help understand the properties of two-dimensional materials: (1) DOI: 10.1016/j.mattod.2024.08.024; (2) 10.1016/j.cplett.2023.140824
Author Response
Thank you very much for taking the time to review this manuscript. Please find the detailed responses below and the corresponding revisions/corrections highlighted/in track changes in the re-submitted files. Please find attached a point-point-rebuttal-reply to reviewer 1

Reviewer 2 Report
Comments and Suggestions for Authors
Overall, the submitted manuscript looks more like an interim (laboratory) report of a computational study of the surface effects associated with the h-BCN molecule deposited on the various substrates. To a much lesser extent, this manuscript could be considered at the level of a scientific paper. Unfortunately, such a distinction makes the review process very difficult. There are three main aspects that the authors could consider and implement to sure that the information provided will be valuable to the scientific community.
1) The scientific style and writing of the manuscript need a complete revision. The title is unclear; the abstract is confusing. The Introduction section is overloaded with a large amount of information. The Conclusions section is limited and incomplete; it is unclear what new information or scientific knowledge has been gained in this paper. The main text is difficult to read. As a result, a number of sentences in the manuscript may not be understandable. Some of the figures are printed too small.
2) In the context of the results of computational simulations, the paper does not provide enough information to judge the reliability of the model and the applicability of the model approximations used. In fact, since they are not properly described at all, it is very difficult to understand why they lead to results. In this case, the computational results seem to be not so useful and informative.
3) In the context of the simulation scenario, the paper contains very little chemistry. Although the results provide some insight, following common practice, the manuscript could include a standard description of the computational chemistry results in terms of chemical classification, composition, relaxed atomic (claster) configurations, valence charge transfer, local symmetry, and structural stability. Two points can be highlighted: (i) the presentation of the results does not meet the standards of modeling in computational chemistry, and (ii) the computational part, methods, and supporting documentation are not well described and properly referenced.
Author Response
Thank you very much for taking the time to review this manuscript. Please find the detailed responses below and the corresponding revisions/corrections highlighted/in track changes in the re-submitted files. Please find attached the point-point-rebuttal reply to reviewer 2.

Reviewer 3 Report
Comments and Suggestions for Authors
Alsad et al discuss the proximity effect of optically active h-BCN structure. The topic is of fundamental interest to community and authors have presented the work clearly. However, my suggestion is to further improve the article through a minor revision. My suggestions are below.
1. Authors provided a clear introduction/aim for this study but the discussions did not clearly emphasized the outcome.
2. Figure 1 and 2 needs corrections in scales/labels.
3. Do authors performed calculation with different supercell/periodicity an dif yes, how does this will affect the outcome.
4. Can authors comments on the effect of stacking arrangement between WSe2 and hBCN.
5. Most of the places "2" in "WSe2" needs to in subscript.
Comments on the Quality of English LanguageEnglish is perfectly fine for publication. Some minor technical corrections may be required.
Author Response
Thank you very much for taking the time to review this manuscript. Please find the detailed responses below and the corresponding revisions/corrections highlighted/in track changes in the re-submitted files. Please find attached the point-point-rebuttal reply to reviewer-3.
